# The Organizational Aspect of Human Resource Management as a Determinant of the Potential of Polish Hospitals to Manage Medical Professionals in Healthcare 4.0

**Beata Buchelt [1,]\*** , **Aldona Frączkiewicz-Wronka [2]** and **Małgorzata Dobrowolska [3]**

1   Department of Human Capital Management, Cracow University of Economics, 31-510 Krakow, Poland
2   Department of Public Management, University of Economics in Katowice, 40-287 Katowice, Poland;
    afw@ue.katowice.pl
3   Institute of Education and Communication Research, Silesian University of Technology,
    44-100 Gliwice, Poland; malgorzata.dobrowolska@polsl.pl
*   Correspondence: beata.buchelt@uek.krakow.pl

**Abstract:** Industry 4.0 solutions have penetrated the healthcare sector, thus creating challenges that healthcare entities should meet. For this, a proper relationship between human resource management (HRM) within healthcare entities and Healthcare 4.0 is needed. In addition, organizations mainly focus on HRM practices, yet organizational issues are overlooked. In this context, the aim of the article was to analyze and evaluate the involvement and roles of key HRM actors, such as line managers and human resource (HR) specialists (HR departments) within strategic healthcare entities, namely hospitals. It was also important to identify the potential of hospitals to meet the requirements of Healthcare 4.0. A study was performed on a group of 285 Polish hospitals. Five respondents were recruited from each hospital. The total population amounted to 1425 interviewees. Due to the complexity of the research, it was outsourced. The results of the analysis identified that hospitals largely engage line managers in medical personnel management. However, a lack of managerial competence may become a major barrier in coping with challenges created by Healthcare 4.0. Organizational solutions do not support the strategic role of HR specialists; their anchoring in the organizational structures limits the possibility to support the changes required for the transformation towards Healthcare 4.0.

**Keywords:** Healthcare 4.0; human resource management; HR actors; hybrid managers; HR specialists; hospitals

## 1. Introduction

The revolution of the manufacturing sector towards Industry 4.0 started a while ago. It gradually affected various branches of the economy and also entered the service sector. Despite the fact that the concept of Industry 4.0 ventured into the new ground of healthcare [1], its original assumptions remained unchanged. In addition, Industry 4.0 allows products, machines, components, individuals, and systems to create a smart network [2–4], thus enabling it to integrate cyberphysical systems (CPS) and perform more quickly by linking information and physical memory to the smart network. As a result of this smart network, products and services can be delivered both more quickly and more effectively [5–7]. Healthcare 4.0 is a collective term for data-driven digital health technologies, such as smart health, mobile health, wireless health, e-health, online health, medical IT, telemedicine, digital medicine, health informatics, pervasive health, and the health information system. It describes the

digital frontiers and disruptive innovation in the healthcare sector that are creating new business models and value networks [8]. The new approach to the delivery of care is expected to enhance the quality and effectiveness of healthcare services, paving the way for a more direct relationship between patients and providers [9].

Healthcare 4.0, artificial intelligence, and digitalization revolutionized the design and the delivery of care. A literature review enabled the identification of the following set of implications of implementing Healthcare 4.0:

- Strategy creation and reconstruction of organizational structures [10];
- The judicious management of healthcare resources aimed at greater efficiency in the delivery of care and improved quality of healthcare services [11];
- Better allocation of resources [12];
- A new patient-centered philosophy of care [13] resulting in co-production and the co-creation of value in healthcare services [14];
- Increasing the ability to monitor and manage the flow of activities performed by healthcare professionals [15];
- Personalized medicine [16];
- The permanent development of knowledge within the healthcare sector [9].

In-depth analysis of Healthcare 4.0 implications reveals the fact that the effects of the revolution are progressing in two different ways—in both medicine itself and in the management of healthcare organizations; this is especially true in hospitals, which create the foundation of the healthcare system. Both of these require an appropriate level of human resource management [17]. Additionally, medical professionals are performing in increasingly complex and often hostile environments, which influence them both directly and indirectly. It is mainly doctors and nurses who are facing the challenges arising from Healthcare 4.0, such as the need for the continual improvement of medical competences, as well as the improvement of competences connected with greater interaction with patients and digital strategies guided by artificial intelligence. Consequently, a requirement is created for healthcare providers to properly handle the relationship between human resource management (HRM) and Healthcare 4.0.

Considering the organizational challenges emerging from Healthcare 4.0 towards human resource management, it should be noted that activities should be taken in two ways: Throughout various human resource management practices and, simultaneously, by optimal solutions that consider the organizational aspects of HRM; in particular, a set of roles and the engagement of critical human resource (HR) actors, such as HR specialists and line managers. Due to the fact that the vast majority of research on HRM in Healthcare 4.0 focuses on processes, methods, and tools for the management of medical professionals, this paper is devoted to the organizational aspects of their management. This orientation results from the fact that thematic literature indicates the relationship between the activities of line managers and HR specialists and the implementation of effective HRM practices [18]. Therefore, the purpose of this paper is to analyze and evaluate the involvement and roles of the key HRM actors, such as line managers and HR specialists, within hospitals. Furthermore, conclusions drawn from the analysis will enable the identification of the potential of the hospitals to effectively meet the requirements of Healthcare 4.0.

## 2. The Main Actors of Human Resource Management in Hospitals: Literature Review

The literature mentions five HRM actors: Top management, line managers (operational managers), employees (often represented by labor unions), HR departments or specialists (depending mostly on the size of the company in question), and, finally, external consultants. The participation of these entities in HRM varies—some are more involved and others less so. HRM actors whose engagement is more evident than others are line managers and HR specialists or HR departments (organizational solution depends on various factors; for example, size of an organization). In actuality, with their activities driven by their roles, these two actors determine the potential of hospitals to manage medical

personnel effectively in accordance with environmental changes, i.e., the development of Healthcare 4.0. Although both of these actors are strategic for healthcare entities to manage medical personnel, their scopes of activity differ. In addition, line managers within hospitals attract the attention of the majority of researchers who investigate the organizational aspect of HRM in healthcare. This is due to the fact that they create a very unique cohort, namely hybrid managers (HMs) [19,20], defined as professionals who perform formal managerial roles within an organization [21]. Although HMs have performed their activities for years now, a changing model of healthcare performance increases the organizational demands placed on the cohort. In addition, a contemporary set of roles performed by operational managers in hospitals comprises [22]: Innovators, brokers, producers, directors, coordinators, monitors, facilitators, and mentors (see Table 1).

**Table 1.** Roles of line managers in hospitals.

| Role Performed by Hybrid Manager (HM) | Characteristics of the Role |
|---|---|
| Innovator | This role considers contemporary determinants of healthcare organizations' performance, namely life quality and life expectancy. Due to the fact that the healthcare sector is knowledge-driven, HMs as innovators need to be entrepreneurial, highly motivated, and responsible. These managers need to implement innovations in the context of daily operations. |
| Broker | This role is about ensuring the continuity of the organization and obtaining external resources. The strength of the negotiation position of the HM in performing this role comes not only from the organization, but also from the broker themselves. In addition, HMs' power also comes from their specialized expertise. This power helps them to influence the flow of resources. Furthermore, this role is determined by an HM's capability to build alliances and partnerships with other organizations, e.g., academic centers. |
| Producer | This role refers to the creation of a productive work environment, optimizing performance, and managing stress and time. |
| Director | This role involves the communication of a vision and the operationalization of goals and objectives as defined in the organization strategy. |
| Coordinator | This role is connected with the maintenance of the team structure and schedules and the coordination of staff efforts. |
| Monitor | This role is about managing information effectively, problem-solving, and decision-making in order to effectively manage risky core processes. |
| Facilitator | This role involves the ability to build and manage teams, especially therapeutic teams, which are constructed by a group of various professionals. It also requires assurance of participative decision-making and the ability to manage conflicts via collaborative approaches. |
| Mentor | This role of the HM is associated with understanding not only others, but also themselves. It requires effective communication and the development of employees through delegation and participation. |

Source: Self-elaboration based on [22–24].

As highlighted in Table 1, hybrid managers are expected to undertake not only varied, but also complex roles and, as a result, perform numerous activities. Years of studies focused on the activities of line managers in hospitals have resulted in the following findings:

- Shipton et al. [25] stressed that in hospitals, the adoption of strategic partners and commitment-creator managers by hospitals apparently fosters affective commitment.
- Sheridan, Vredenburgh et al. [26] observed that when hybrid managers display leadership behaviors, the work efficiency of their subordinates increases.
- Hybrid doctor–managers are often autocratic [19], yet this does not affect their openness to change [27].

The HR department is the second HRM actor that is permanently engaged in medical personnel management. Its main activities should be concentrated on support of line managers and employees;

thus, it should deliver adequate resources and tools and also administrate the HRM process [18]. In order to achieve this aim, HR departments should be highly placed in the organizational structures of hospitals. Moreover, a person responsible for medical personnel management issues should be a part of the board of directors. There is a set of HR department roles within hospitals (see Table 2): Strategic partner, change agent, administrative expert, and employee champion [28].

**Table 2.** Roles of human resource (HR) departments.

| Role Performed in HR Department | Role Characteristic |
|---|---|
| Strategic partner | This role requires being able to create a human resource management (HRM) strategy that is an integral part of the organizational strategy. |
| Administrative expert | This role is concerned with carrying out administrative activities in such a way that the focus is not only on the performance of activities, but also on increasing their efficiency and reducing their cost-effectiveness through better organization of the tasks being carried out. |
| Change agent | This role is associated with preparing and implementing changes within the organization. It assures that organizational culture is considered when changes are being planned and convinces employees to embrace changes and innovations. |
| Employee champion | A person in this role listens and responds to an employee's personal needs. They provide a working environment in which employees are motivated, productive, and happy at work. |

Source: Self-elaboration based on [28–30].

Khatri, Wells et al. [29] noted that in order for the HR department in a hospital to fulfill its roles, the following conditions must be met:

- There must be a competent HR manager and a board of directors that is aware of the importance of human resources in hospitals.
- HR specialists must be of the required level of competence.
- There must be an increased status of HRM issues.
- The HR department must permanently improve its competences.
- There must be an integrated IT system supporting the management of employees from administrative to strategic issues.

Unfortunately, the problem is that, in practice, there is a clear shortage of specialists with specific competences related to human resource management in hospitals [29]. Their quantitative and qualitative appreciation is vulnerable to the underestimation by top management of their key role in the organization, and also the strong position of medical professions. With regard to the latter factor, it is even suggested that HR specialists have failed by weakening the status of key medical professions [31].

## 3. Material and Methods

The data required for attainment of the aims of the study were gathered from 285 Polish hospitals assuming a confidence level of 0.95. The survey primarily targeted hospital directors due to the fact that, in practice, these managers remain involved in HRM activities in hospitals to a large extent. There were also four other groups of respondents participating in the study. The multi-perspective research design was deemed important because of the possible overstatement of answers—this is known as officialization [32]. All respondents remained anonymous. Research was performed operationally by a specialized research company with experience in the healthcare sector. The study took place in 2017. The research company contacted the directors first. Directors familiarized themselves with the thematic scope of the survey and made decisions regarding whether they would personally respond or delegate participation. Most commonly, they identified subordinates responsible for HRM. The

group of respondents consisted of: 57 directors (20%), 138 HR managers (48%), 83 HR specialists (29%), and 7 other respondents (2%), e.g., public relations specialists. Hospitals were asked to forward the survey to other groups of respondents, such as physicians and nurses serving as ward managers and line managers (also named hybrid managers), as well as individual physicians and nurses. There was a total of 1425 respondents from the 285 hospitals who participated in the survey. Worth noting is that there were three major versions of the survey covering the same scope of questions. The main difference was that the surveys for HMs and employees were shorter than that used for directors, and they were focused only on key HRM issues. Moreover, the surveys were adjusted to the type of respondent where necessary so they would know that they were asked about a precise group of professionals (physicians or nurses).

The survey was a part of a larger project entitled Human Capital Management in Hospitals, financed by Polish National Science Center (DEC–2013/11/B/HS4/01062). It was designed based on an extensive literature review (i.e., [28–31]) and discussions among researchers. The scope of the survey was broader than the aim of this paper. In addition, the whole spectrum of questions was aimed at the identification of the stage of development of personnel function in Polish hospitals. For the purpose of the paper, one out of a set of questions was chosen that considered aspects connected with the roles and scope of the activities of the main HRM actors—line managers (HMs) and HR specialists (HR department). A set of five questions analyzed for the purpose of the paper consisted of the following:

(1) A question concerning the extent of HM involvement in the medical personnel management practices (five-point Likert scale).
(2) A question on the activities aimed at HM management competence development—this was included in the 'Director' and 'HM' surveys.
(3) A question to evaluate the HRM competences of HMs—this was included in the 'Physician' and 'Nurse' surveys.
(4) A question regarding organizational solutions for HR specialists/HR departments—this was included in the 'Director' and 'HM' surveys.
(5) A question on the roles of HR specialists/HR departments (five-point Likert scale).

Data were analyzed using Statistica 13. Distribution analysis was applied to the questions. The different groups of respondents had the following characteristics (see Table 3):

- Cohort 'Directors': Most of the respondents were between 45 and 54 years old; 80.4% were female and had worked for 10–19 years.
- Cohort 'Hybrid Doctor–Managers': Most of the respondents were between 45 and 54 years old; 57.7% were male and had worked for 20–29 years.
- Cohort 'Hybrid Nurse–Managers': Around the same proportions of respondents belonged to age groups 35–44 and 45–54 years (approximately 33% per age group), all were female, and most of them had worked for 20–29 years.
- Cohort 'Physicians': 95.5% of respondents belonged to the 35–54 age group, 57% were female, and 51.2% had worked for 10–19 years.
- Cohort 'Nurses': 95.5% of respondents belonged to the 35–54 age group, all were female, and 42.5% had worked for 10–19 years.

**Table 3.** Characteristics of the researched populations (%).

|  | Directors | HDM * | HNM ** | Physicians | Nurses |
|---|---|---|---|---|---|
| | | | Age | | |
| No response | 7.7 | 0 | 0 | 0 | 0 |
| 25–34 | 1.4 | 0 | 4.2 | 0.4 | 1.8 |
| 35–44 | 26.3 | 16.8 | 33.3 | 45.3 | 46.0 |
| 45–54 | 38.9 | 43.9 | 35.8 | 50.2 | 49.5 |
| 55–64 | 24.9 | 38.2 | 26.7 | 3.9 | 2.8 |
| 65–74 | 0.7 | 1.1 | 0.0 | 0.4 | 0.0 |
| | | | Gender | | |
| Female | 80.4 | 42.5 | 100 | 57.2 | 100 |
| Male | 19.6 | 57.7 | 0 | 42.8 | 0 |
| No response | 1.4 | 0 | 0 | 0 | 0 |
| | | | Seniority | | |
| 0–9 | 23.9 | 4.2 | 14.0 | 26.7 | 30.9 |
| 10–19 | 29.8 | 32.3 | 28.4 | 51.2 | 42.5 |
| 20–29 | 24.9 | 45.3 | 41.1 | 21.4 | 25.6 |
| 30–39 | 16.5 | 16.5 | 15.8 | 0.7 | 1.1 |
| 40–49 | 3.2 | 1.1 | 0.7 | 0 | 0 |
| 50–59 | 0.4 | 0.7 | 0.0 | 0 | 0 |

* Hybrid Doctor–Managers—HDM; ** Hybrid Nurse–Managers—HNM

## 4. Results

Line managers should regularly be involved in the management of their teams and subordinates. This involvement can be detected throughout their activities. In general, the majority of respondents declared that HMs are regularly engaged in various HRM activities. When analyzing the data in Table 4, the following remarks can be made:

- 'Directors' declared that line managers are mostly engaged in performance appraisal, team management, conflict management, and training. However, they are actually not present in career management, recruitment, and selection.
- 'Hybrid Doctor–Managers' (HDMs) declared that they are highly involved in team management, performance appraisal, introduction, and staffing. They also highlighted dismissal of employees, career management, and the implementation of changes within HR functioning as practices in which they do not participate.
- 'Hybrid Nurse–Managers' (HNMs) declared that they are highly involved in team management, training, introduction, and HR planning. However, they are not involved in the dismissal of employees, the implementation of changes within HR functioning, or remuneration.
- 'Physicians' declared that HDMs are highly involved in team management, conflict management, and performance appraisal. However, they are either rarely or not at all engaged in career management, remuneration, or the implementation of changes within HR functioning.
- 'Nurses' declared that HNMs are engaged in conflict management, team management, and performance appraisal, but are rarely involved in remuneration, career management, or the dismissal of employees.

**Table 4.** Involvement of HMs in medical personnel management practices (%).

| MPMP * | 1 | 2 | 3 | 4 | 5 | 1 | 2 | 3 | 4 | 5 | 1 | 2 | 3 | 4 | 5 | 1 | 2 | 3 | 4 | 5 | 1 | 2 | 3 | 4 | 5 |
|---|---|---|---|---|---|---|---|---|---|---|---|---|---|---|---|---|---|---|---|---|---|---|---|---|---|
| | DIRECTORS | | | | | HDM | | | | | HNM | | | | | PHYSICIANS | | | | | NURSES | | | | |
| HR planning | 0.4 | 2.8 | 7.7 | 70.2 | 18.9 | 1.8 | 8.8 | 2.1 | 69.8 | 17.5 | 2.8 | 4.6 | 2.5 | 65.3 | 24.9 | 4.9 | 1.1 | 9.5 | 58.2 | 26.3 | 3.5 | 1.4 | 10.5 | 59.3 | 25.3 |
| Recruitment | 0 | 0 | 14 | 72.3 | 13.7 | 4.9 | 2.1 | 6.3 | 67.0 | 19.6 | 3.5 | 4.2 | 4.9 | 63.9 | 23.5 | 3.2 | 3.2 | 10.2 | 55.8 | 27.7 | 4.6 | 2.8 | 10.9 | 55.8 | 26.0 |
| Selection | 0 | 0 | 14 | 67.4 | 18.6 | 1.8 | 3.2 | 5.3 | 63.9 | 26.0 | 3.2 | 3.5 | 3.9 | 66.0 | 23.5 | 3.5 | 2.8 | 9.1 | 53 | 31.6 | 3.9 | 2.5 | 11.6 | 51.9 | 29.8 |
| Introduction | 0.4 | 0 | 10.9 | 60.0 | 28.8 | 4.2 | 1.1 | 3.9 | 59.3 | 31.6 | 2.5 | 1.4 | 2.8 | 55.8 | 37.5 | 3.5 | 2.5 | 10.2 | 47.4 | 36.1 | 5.3 | 3.2 | 6.0 | 48.4 | 36.8 |
| Team mgmt. | 0 | 0 | 8.8 | 54.7 | 36.5 | 1.4 | 0.7 | 1.1 | 60.0 | 36.8 | 1.1 | 1.4 | 1.4 | 54.7 | 41.4 | 2.8 | 0 | 9.1 | 44.6 | 43.5 | 1.8 | 0.4 | 8.1 | 48.1 | 41.8 |
| Performance appraisal | 0 | 0.4 | 6.0 | 51.2 | 42.5 | 1.4 | 0.4 | 7 | 53.3 | 37.9 | 1.1 | 2.1 | 6.3 | 53.3 | 37.2 | 0 | 4.9 | 9.5 | 45.3 | 40.4 | 0 | 1.4 | 9.1 | 45.3 | 44.2 |
| Staffing | 0.4 | 0 | 5.6 | 55.1 | 38.9 | 1.8 | 0.7 | 6.7 | 58.9 | 31.9 | 0.7 | 1.8 | 11.9 | 55.8 | 29.8 | 2.5 | 0.7 | 9.8 | 48.1 | 38.9 | 1.4 | 0.4 | 12.6 | 47 | 38.6 |
| Training | 0 | 0 | 8 | 54.4 | 36.8 | 1.4 | 3.9 | 2.1 | 58.9 | 33.7 | 1.1 | 0.7 | 2.5 | 60.7 | 35.1 | 2.1 | 4.2 | 10.9 | 41.4 | 41.4 | 2.5 | 4.2 | 8.8 | 47 | 37.2 |
| Career mgmt. | 0.4 | 5.3 | 8.4 | 52.3 | 33.7 | 7.7 | 8.8 | 6.3 | 58.9 | 18.2 | 5.3 | 9.5 | 4.2 | 58.6 | 22.5 | 4.6 | 2.1 | 17.5 | 49.1 | 26.7 | 4.2 | 2.5 | 17.2 | 49.5 | 26.3 |
| Remuneration | 0 | 2 | 11.2 | 60 | 26.0 | 10.2 | 7.0 | 3.5 | 64.2 | 15.1 | 14 | 6.3 | 3.2 | 59.6 | 16.8 | 9.1 | 0 | 15.1 | 47.0 | 28.8 | 6 | 0 | 19.3 | 47 | 27.7 |
| Conflict management | 0 | 0 | 8.8 | 56.5 | 34.7 | 2.1 | 5.3 | 7.7 | 56.1 | 28.8 | 2.5 | 4.9 | 7.4 | 48.8 | 36.5 | 1.4 | 3.9 | 8.1 | 50.5 | 35.8 | 0.7 | 3.5 | 4.9 | 53 | 37.9 |
| Dismissal of employees | 0.4 | 2.5 | 8.4 | 57.2 | 31.6 | 7.7 | 10.5 | 6.0 | 58.9 | 16.8 | 9.5 | 11.6 | 9.1 | 57.2 | 12.6 | 3.9 | 2.5 | 13.7 | 46.3 | 33.7 | 5.6 | 4.2 | 14.0 | 50.2 | 26.0 |
| Implementation of changes within HR function | 0.4 | 5.6 | 5.6 | 59.6 | 28.8 | 7.0 | 5.6 | 10.2 | 59.3 | 17.9 | 8.4 | 3.2 | 11.6 | 58.2 | 18.6 | 1.8 | 1.8 | 17.2 | 47.4 | 31.9 | 2.8 | 0 | 18.2 | 47.4 | 31.6 |

* MPMP—medical personnel management practices; 1—Does not participate/I do not participate; 2—Participate very rarely/I participate very rarely; 3—Participate rarely/I participate rarely; 4—Often participate/I often participate; 5—Participate regularly/I participate regularly.

Interesting conclusions came from the Kruskal–Wallis analysis (see Table 5). In the cases of four out of thirteen practices, a significant difference in the shape of the distribution between the positions can be identified. Most commonly, the differences concern 'Directors' and HMs. However, there are also significant differences identified between HDMs and 'Physicians' or 'Nurses', as well as between HNMs and 'Nurses' or 'Physicians'. The analysis of the data in Table 4 in comparison with the data in Table 5 revealed that the higher the position of the respondent in the organizational hierarchy, the greater involvement of line managers in medical personnel management is declared.

**Table 5.** MPMP—significant differences of the answer distributions among respondents.

| MPMP ** | Kruskal-Wallis Statistic | $p$ | The Significant Differences in Distribution of the Cohorts' Answers (Individual $p$) |
|---|---|---|---|
| Career management | 22.7151 | 0.0001 | Director—HDM (0.0369)<br>Director—HNM (0.0002)<br>Director—Physicians (0.0864)<br>Director—Nurses (0.0859) |
| Remuneration | 22.0312 | 0.0002 | Director—HDM (0.0092)<br>Director—HNM (0.0033) |
| Dismissal of employees | 67.2169 | <0.0001 | Director—HDM (<0.0001)<br>Director—HNM (<0.0001)<br>Director—Nurses (0.0319)<br>HDM—Physicians (0.0016)<br>HNM—Physicians (<0.0001)<br>HNM—Nurses (0.0053) |
| Implementation of changes within HR function | 29.8615 | <0.0001 | Director—HDM (0.0024)<br>Director—HNM (0.0033)<br>HDM—Physicians (0.0276)<br>HDM—Nurses (0.0381)<br>HNM—Physicians (0.0359)<br>HNM—Nurses (0.0491) |

* MPMP—medical personnel management practices; ** Only practices where the significant differences were identified are taken into consideration; $p < 0.1$ is indicated in italics.

Engagement in activities related to the management of subordinates requires the permanent improvement of knowledge and skills in this area. The investigation dedicated to this issue was performed doubly. 'Directors', 'HDMs', and 'HNMs' were asked about the availability of the development activities (see Table 6). 'Physicians' and 'Nurses' were asked to appraise their supervisors' managerial competences (see Table 7). Training availability was rated highly by 'Directors'. Of the 'Directors', 70.2% stated that such development activities are performed regularly. By contrast, 9.1% of 'HDMs' stated that such training is not organized in their hospital at all, and 10.9% declared that it is performed rarely. Both 'HDMs' and 'HNMs' declared that the activities are performed sometimes. The identified variations are confirmed by the Kruskal–Wallis analysis. In addition, for Directors—HDMs and for Directors—HNMs, $p < 0.0001$. 'Physicians' appraised the managerial competences of their supervisors more critically than 'Nurses'. However, the U-Mann Whitney test did not expose significant differences in the responses of 'Physicians' and 'Nurses'.

**Table 6.** Activities aimed at updating the HM knowledge and skills in the field of medical personnel management.

| Response Options | Directors | HDM | HNM |
|---|---|---|---|
| DAs * are not performed | 0.0% | 9.1% | 6.3% |
| DAs are performed rarely | 3.2% | 10.9% | 12.6% |
| DAs are performed sometimes | 2.8% | 50.2% | 46.3% |
| DAs are performed moderately regularly | 23.9% | 24.6% | 26.7% |
| DAs are performed very regularly | 70.2% | 5.3% | 8.1% |

DAs—development activities.

**Table 7.** Evaluation of HM personnel management competences by subordinates.

| Response Options | Physicians | Nurses |
|---|---|---|
| They are very low | 0.0% | 0.0% |
| They are low | 0.0% | 0.0% |
| They are neither low nor high | 44.9% | 37.5% |
| They are high | 37.5% | 42.8% |
| They are very high | 17.5% | 19.6% |

Three groups of respondents were asked about organizational solutions concerning the HR department. Declarations of 46.7% of 'Directors' placed the HR department on a tactical level, and 29.8% placed it on the strategic (the highest) level. By contrast, only a marginal group of HMs declared that the department is on the strategic level. The majority of the respondents indicated that there are only HR specialists, rather than entire HR departments, employed within hospitals (see Table 8). The Kruskal–Wallis analysis confirmed the observation. The distributions of answers for 'Directors'—'HDM' and 'Directors'—'HNM' are significant; for both, $p < 0.0001$.

**Table 8.** HR organizational solutions concerning HR specialists or HR departments with the hospitals' organizational structures.

| Response Options | Directors | HDM | HNM |
|---|---|---|---|
| There is no position in the hospital where an employee deals with personnel issues. | 0.0% | 0.0% | 0.0% |
| A specialist is employed in the hospital who, among other things, deals with personnel issues. | 2.8% | 9.1% | 8.4% |
| A specialist is employed in the hospital who only deals with personnel issues. | 20.7% | 43.9% | 42.5% |
| The organizational structure of the hospital has a department for personnel matters managed by a mid-level manager. | 46.7% | 34.7% | 31.6% |
| There is an HR department in the hospital, which is managed by a top manager (a member of a board of directors). | 29.8% | 12.3% | 17.5% |

All respondents were asked to share their opinions about roles taken by HR department (see Table 9). 'Directors' were the ones who clearly declared various roles of the department; 99.7% of them 'agreed' or 'strongly agreed' that the HR department plays an administrative role. HNMs also clearly identified this role of the department. 'Participation in the process of shaping interpersonal relations at the hospital' was strongly evaluated by 'Directors'. Strategic partnership was highly valued not only by 'Directors', but also by HNMs. 'Participates in the processes of organizational change, acting as a so-called change agent' scored strongly among HDMs and 'Directors'. Similarly, the respondents appraised the initiation of change undertaken by the HR department. Finally, the provision of effective HRM methods and tools was evaluated highly by 'Directors' and comparatively low by HDMs. Here, as previously, the Kruskal–Wallis test (see Table 10) was also carried out. It revealed interesting points concerning the perception of the various roles by the groups of respondents. These are:

- The administrative role of HR specialists was generally appraised as high by 'Directors' and 'HNMs'.
- The role connected with shaping interpersonal relations was appraised the highest by 'Directors'. Interestingly, 'Physicians' and 'Nurses' appraised this role higher than HMs.
- Strategic partnership was appraised differently only by 'Physicians'.
- 'Directors' appraised the role of the change agent strongly. Their answers differed from those of other cohorts.
- HNMs obviously perceived HR specialists as initiators of the projects within hospitals. By contrast, HDMs appraised the role the lowest.
- Answers of 'Directors' differed from those of other cohorts in relation to the provision of effective HRM tools and methods. 'Physicians' appraised the role the lowest.

**Table 9.** HR specialist/HR department roles.

| Scale: | Strongly Disagree | Disagree | Neither Agree nor Disagree | Agree | Strongly Agree |
|---|---|---|---|---|---|
| It handles administrative matters of employees | | | | | |
| Directors | 0.0% | 0.0% | 0.4% | 79.3% | 20.4% |
| HDM | 6.7% | 2.5% | 6.3% | 67.4% | 17.2% |
| HNM | 0.0% | 0.0% | 0.4% | 65.6% | 34.0% |
| Physicians | 2.1% | 3.5% | 9.1% | 67.7% | 17.5% |
| Nurses | 2.5% | 1.4% | 10.5% | 61.8% | 23.9% |
| Participates in the process of shaping interpersonal relations at the hospital | | | | | |
| Directors | 0.0% | 8.4% | 8.4% | 65.6% | 17.5% |
| HDM | 5.3% | 6.7% | 16.1% | 64.6% | 7.4% |
| HNM | 7.7% | 6.3% | 12.3% | 66.7% | 7.0% |
| Physicians | 7.4% | 4.6% | 18.2% | 56.8% | 12.6% |
| Nurses | 7.4% | 3.2% | 22.1% | 57.2% | 10.2% |
| It is a strategic partner in solving important problems within the hospital | | | | | |
| Directors | 0.0% | 8.8% | 2.8% | 77.5% | 10.9% |
| HDM | 1.8% | 10.5% | 11.9% | 64.6% | 11.2% |
| HNM | 2.8% | 4.9% | 11.9% | 70.2% | 10.2% |
| Physicians | 6.7% | 3.9% | 20.7% | 55.4% | 13.3% |
| Nurses | 6.3% | 3.2% | 18.2% | 59.3% | 12.6% |
| Participates in the processes of organizational change, acting as a so-called change agent | | | | | |
| Directors | 0.4% | 4.9% | 5.6% | 62.5% | 26.7% |
| HDM | 0.7% | 13.3% | 15.4% | 58.9% | 11.6% |
| HNM | 1.8% | 10.2% | 10.2% | 66.3% | 11.6% |
| Physicians | 4.6% | 5.3% | 19.3% | 51.9% | 18.2% |
| Nurses | 7.7% | 2.5% | 20.7% | 50.5% | 18.6% |
| It is the initiator (leader) of important projects in the hospital | | | | | |
| Directors | 3.2% | 8.1% | 5.3% | 54.4% | 29.1% |
| HDM | 12.3% | 18.6% | 22.1% | 31.6% | 15.4% |
| HNM | 0.7% | 2.1% | 2.1% | 80.7% | 14.4% |
| Physicians | 7.7% | 2.5% | 20.7% | 50.5% | 18.6% |
| Nurses | 4.9% | 2.5% | 21.4% | 50.2% | 21.1% |
| It provides effective methods and tools for managing medical personnel. | | | | | |
| Directors | 2.5% | 6.0% | 0.0% | 59.6% | 31.9% |
| HDM | 2.8% | 12.3% | 17.9% | 53.7% | 13.3% |
| HNM | 1.8% | 11.9% | 11.6% | 60.7% | 14.0% |
| Physicians | 6.3% | 3.9% | 14.4% | 52.3% | 23.2% |
| Nurses | 9.1% | 2.5% | 18.2% | 52.6% | 17.5% |

**Table 10.** HR specialist roles—significant differences in the answer distributions among respondents.

| HR Specialist Roles | Kruskal–Wallis Statistic | *p* | The Significant Differences in Distribution of the Cohorts' Answers (Individual *p*) |
|---|---|---|---|
| Handles administrative matters of employees. | 64.6973 | <0.0001 | Directors—HDM (0.0213) Directors—Physicians (0.0490) HDM—HNM (<0.0001) HNM—Nurses (0.0004) HNM—Physicians (<0.0001) |
| Participates in the process of shaping interpersonal relations at the hospital. | 27.768 | <0.0001 | Directors—HDM (0.0030) Directors—HNM (0.0040) Directors—Physicians (0.0128) Directors—Nurses (0.0009) |
| Is a strategic partner in solving important problems within the hospital. | 15.2560 | 0.0042 | Directors—Physicians (0.0266) |
| Participates in the processes of organizational change, acting as a so-called change agent. | 47.1213 | <0.0001 | Directors—HDM (<0.0001) Directors—HNM (0.0001) Directors—Physicians (0.0001) Directors—Nurses (<0.0001) |
| Is the initiator (leader) of important projects in the hospital. | 114.6863 | <0.0001 | Directors—HDM (<0.0001) Directors—Physicians (<0.0001) Directors—Nurses (0.0483) HDM—HNM (<0.0001) HDM—Physicians (0.0019) HDM—Nurses (<0.0001) HNM—Physicians (<0.0001) |
| Provides effective methods and tools for managing medical personnel. | 66.2373 | <0.0001 | Directors—HDM (<0.0001) Directors—HNM (<0.0001) Directors—Nurses (<0.0001) Directors—Physicians (0.0014) HDM—Physicians (0.0450) |

## 5. Discussion and Conclusions

Along with the development of Industry 4.0, Healthcare 4.0 has evolved, creating an unknown, turbulent, yet unique environment for healthcare entities. It has dramatically revolutionized the design and delivery of care, and has also created implications for the performance of healthcare entities. As a consequence, it creates the requirement for the providers to properly manage medical professionals [17]. In this context, effective HRM practices become a necessity for the healthcare entities to be successful. A major condition for HRM practices to be effective is the proper engagement of line managers and HR specialists. This is due to the fact that they fulfill the architecture of personnel function within the organizations and create organizational solutions that support HRM practices [33]. For this reason, the aim of this paper was to analyze and evaluate HRM actors' roles and engagement within the most important healthcare entities—hospitals. Per se, the identification of the state of the art of line managers' and HR specialists' roles and involvement determines the potential of the hospitals to effectively meet Healthcare 4.0's requirements. This is because HRM organizational solutions either support or constitute barriers for effective human resource management practices.

In the optimal model of HM engagement within modern organizations, these line managers are responsible for managing medical personnel on an ongoing basis [34]. The spectrum of desired HM activities is broad, ranging from planning, organizing work, acquiring new employees, and assessing the work of subordinates, to conflict resolution and implementing changes. Thus, HM involvement results in higher performance of medical personnel [26], which may support the implementation of

innovation. The research confirmed that HMs' operational involvement in human resource management is high in Polish hospitals. The responses of individual groups of respondents in relation to the vast majority of HRM practices overlapped. However, the Kruskal–Wallis analysis identified differences in responses between certain groups of respondents in relation to such practices as: Career management, remuneration, dismissal of employees, and implementation of changes within HR function. Identifying the reasons for these discrepancies, it should be emphasized that the comparison of the results of the indicated statistical test with data presenting the percentage of responses of individual cohorts of respondents indicates a general tendency of the response dependence on the type of cohort. HM involvement was rated the highest by 'Directors' and the lowest by 'Physicians' and 'Nurses'. The identified discrepancies can be supported by the results of diversified research. Not all of these can be explained. In addition, the different positions of respondents towards the involvement of line managers in career management are explained by the fact that individual personal development, which is the domain of these strong medical professions, is controlled by doctors and nurses [18]. The difference in the responses regarding remuneration may be caused by the fact that not only the remuneration strategy, but also operational practices connected with the compensation are centralized or even regulated externally by industry-normative acts [35]. Additionally, this is especially true for Poland.

Further in-depth analysis of the responses reveals that HMs are largely involved in their subordinates' management. HDMs are involved mostly in performance appraisal, team management, and conflict management. Such a conclusion recalls the results of various studies focused on challenges connected with performance appraisal implementation within healthcare entities, during which the dual role played by HDMs [36] is reviewed. Researchers noticed that in modern healthcare entities, performance appraisal is a standard HRM practice [37]. However, its implementation can lead to conflicts arising on the basis of the hybrid nature of line managers.

Identification of the level of HM involvement in human resource management was accompanied by an attempt to assess the development of adequate competences and their assessment from the point of view of subordinates. The obtained results partly confirmed the results of research performed by Denis and van Gestel [20] and Sanford [38]. It was found that the management competences of HMs are thoroughly assessed by their subordinates. Although HMs positively assess the availability of HRM-related training, their effectiveness may be limited by their reluctance to develop in this field [20,37].

This paper has also been aimed at the detection of the roles and engagement of HR specialists (HR departments) in the process of the management of medical professionals. For this reason, the results of two particular questions from the research were analyzed. The first enabled identification of the actual anchoring of HR specialists in the organizational structures of hospitals. The second was aimed at identifying their roles. In addition, it is worth noting that, in these questions, the answers of 'Directors' differed from those of other respondent groups. The phenomenon of officialization is clearly taking place again here. The statements of respondents who did not succumb to this phenomenon show that in the vast majority of the hospitals, solutions relating to organizational structure do not follow the requirements of Healthcare 4.0. Most hospitals employ only one HR specialist who deals with personnel issues. This means that solutions for organizational structure do not follow the requirements of Healthcare 4.0. As a consequence, they are doomed to failure with regard to implementing the organizational strategy. This conclusion is drawn from the research of Khatri et al. [29], who pointed out that a major condition for the HR department to fulfill its roles is to increase its status within healthcare organization. Placing the HR actor on a tactical or operational level does not contribute to strengthening its status in the organization. This conclusion is confirmed by the fact that respondents highlighted the role of administrative experts as the most visible role fulfilled by HR specialists. In addition, the role does not generate added value [33]. The effective HRM in hospitals requires the HR department to be anchored on a strategic level. The present placement of the HR specialists can lead to failure [31] in the implementation of solutions appropriate for Healthcare 4.0.

There is also another interesting conclusion coming from the research. In addition, it can be observed that hybrid nurses managers appraise the roles and engagement of HR specialists higher than the other medical cohorts do. This might be an effect of the higher involvement of nurses in administrative activities. Additionally, in his research, Carney [39] confirmed that nurses, when muddling through work burnout, tend to move to nonmedical positions, i.e., managerial ones. The results of the study also reveal that mostly HDMs, but also 'Physicians', appraise the performance of the HR specialist relatively low compared to other cohorts of respondents. A reason for this phenomenon might be the autonomous character of this profession written into its genotype [40]. This autonomy stands in contradiction with interdisciplinary cooperation, e.g., with nurses [41].

Overall, this paper contributes to the development of management sciences in specific human resource management and healthcare management by bringing in-depth insights into the engagement and roles of the main HRM actors, such as line managers and HR specialists within hospitals. In addition, current studies focus on HRM processes or practices in the context of Healthcare 4.0. The results of the conducted analyses reveal that Polish hospitals are, on the one hand, ready to face Healthcare 4.0, but on the other, some structural solutions have not been able to cope. Additionally, HMs are sufficiently involved in managing subordinates. However, the attention of top management must be focused on the permanent training of line managers and overcoming their resistance to non-specialized training. Much needs to be done in the area of the involvement and roles performed by HR specialists. In general, the organizational structures of hospitals should be reconstructed so that HR specialists begin to play a strategic role in the organizations. Furthermore, the introduced changes should be accompanied by the training of employees who would not be standard HR specialists, but would possess the specialist medical competence to understand the specificity of healthcare organizations. Perhaps the institutions responsible for the functioning of the healthcare system in Poland should consider creating a new profession, such as the HR partner in healthcare, and set a standard for the qualifications that such the employee should meet. This solution should be developed on a macro level. In addition, in Poland, there was a special initiative developed that aimed to set the standards of qualifications not only for the existing, but also the arising medical profession. This initiative is named the Sectoral Competence Council. On a micro level, the position of the HR partner should be anchored in the organizational structure in between boards of directors and HMs. However, this position should be equipped with strong authority and support from top management. The uniqueness of the study and its results come also from the fact that it was performed from various perspectives and visibly exposed the phenomenon of officialization.

Despite the fact that the research was conducted in Poland, it seems obvious that it possesses universal features. This is due to the fact that it relates to aspects that are not recognized, but are relevant to the functioning of healthcare entities in other countries. As pointed out in the literature review, healthcare entities in other countries struggle with the creation of optimal solutions concerning HRM's organizational aspect in the context of Healthcare 4.0. In addition, HMs fail to perform their roles, and their engagement is not as high as expected. Furthermore, HR specialists are unsuccessful in delivering effective HRM solutions, and they also fail to properly perform more advanced roles than administrative roles. This is mainly because they are not able to weaken the strength of strong professions, such as those of physicians most of all, but also nurses.

Nonetheless, this study has opened new lines of future research. The roles of other HR actors could be studied further; for example, top management and labor unions, which are extremely active in public hospitals. Moreover, the relationship between representatives of strong professions and HR specialists could be investigated more deeply. This would be very fruitful from a management perspective to precisely define the set of competences needed for HR specialists to successfully perform their duties and, consequently, their roles. Further studies could be also concentrated on the detection of an optimal HR organizational structure model that could support the implementation of Healthcare 4.0 solutions.

**Author Contributions:** Conceptualization, B.B.; Data curation, B.B. and A.F.-W.; Investigation, B.B. and A.F.-W.; Methodology, B.B.; Project administration, B.B.; Supervision, B.B.; Writing—original draft, M.D.; funding acquisition, M.D. All authors have read and agreed to the published version of the manuscript.

**Funding:** The publication was financed within the framework of the program entitled "Dialogue" introduced by the Minister of Science and Higher Education between 2016 and 2019. The research was supported by a grant (DEC–2013/11/B/HS4/01062) entitled Human Capital Management in Hospitals financed by the Polish National Science Center.

**Conflicts of Interest:** The authors declare no conflict of interest.

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
