# Peer review of "The Organizational Aspect of Human Resource Management as a Determinant of the Potential of Polish Hospitals to Manage Medical Professionals in Healthcare 4.0"

_sustainability, doi:10.3390/su12125118_

Round 1

Reviewer 1 Report

The paper presentes a study on a Polish hospital situation about the ability to face the challenges created by Helthcare 4.0.

The research has been outsourced and it is a part of a larger project (Human Capital Management in Hospitals).
It is not clear how the survery has been conducted. Did the people involved remain anonymous?
Have the answers given remained anonymous?

In order to improve the motivation and the literature I suggest to add these recent references:
- https://doi.org/10.1109/CBI.2019.00025
- https://doi.org/10.1109/JSYST.2020.2968614
- https://doi.org/10.1109/JSEN.2020.2982744
- https://doi.org/10.1109/ICEV.2019.8920566

- The reference to the Abidi and Abidi [11] at row 279 it is not clear. Please explain this.

- In the coclusions the authors say that the research has a universal feature. Is this work applicable to other national health systems? What kind of health system is there in Poland? In the opinion of the authors, could there be a difference between a public and a private system?

Minor:
-Move the HRM ancronym from row 63 to row 19
-Can the authors move the table 4 on a single page? It is very difficult to read it.

Author Response

Dear Sir or Madam,

Thank you for in-depth review of our paper. The suggestion for improvements were very useful and majority of them were engaged in order to improve the paper. Let us refer to the feedback:

S1. It is not clear how the survey has been conducted. Did the people involved remain anonymous? Have the answers given remained anonymous?

Yes, the respondents and their answers were anonymous. Explanation was added in line 143 (prime manuscript).

S2. In order to improve the motivation and the literature I suggest to add these recent references:

https://doi.org/10.1109/CBI.2019.00025

https://doi.org/10.1109/JSYST.2020.2968614

https://doi.org/10.1109/JSEN.2020.2982744

https://doi.org/10.1109/ICEV.2019.8920566

Authors identified the list of papers:

  • Sandkuhl, K. (2019). Putting AI into Context - Method Support for the Introduction of Artificial Intelligence into Organizations. 2019 IEEE 21st Conference on Business Informatics (CBI). doi:10.1109/cbi.2019.00025
  • Ray, P. P., Kumar, N., & Dash, D. (2020). BLWN: Blockchain-Based Lightweight Simplified Payment Verification in IoT-Assisted e-Healthcare. IEEE Systems Journal, 1–12. doi:10.1109/jsyst.2020.2968614
  • Coviello, G., & Avitabile, G. (2020). Multiple Synchronized Inertial Measurement Unit Sensor Boards Platform for Activity Monitoring. IEEE Sensors Journal, 1–1. doi:10.1109/jsen.2020.2982744
  • Perez-Jimenez, M. A., Martinez-Castillo, J., Morales-Gonzalez, E., & Herrera-May, A. L. (2019). A Portable and Wireless System for Monitoring Hand Tremors in Parkinson’s Disease Patients. 2019 IEEE International Conference on Engineering Veracruz (ICEV). doi:10.1109/icev.2019.8920566

In-depth analysis of the papers indicated that, even though they are extremely interesting, their scope does not overlap precisely with our paper area of investigation. 

S3. The reference to the Abidi and Abidi [11] at row 279 it is not clear. Please explain this.

The reference was deleted at the place due to major changes of the part of the paper.

S4. In the conclusions the authors say that the research has a universal feature. Is this work applicable to other national health systems? What kind of health system is there in Poland? In the opinion of the authors, could there be a difference between a public and a private system?

This suggestion, but also suggestions of other reviewers motivated us to redefine conclusions and identify limitations of the study. In addition, due to the fact that the study was performed in Poland, we have indicated that it might be perceived as the study limitation. Polish healthcare system is basically public. However, a share of private entities is growing systematically.

S4. Minor amendments: Move the HRM acronym from row 63 to row 19; Can the authors move the table 4 on a single page? It is very difficult to read it.

The acronym was moved to line 16/17. It is the first line where the term human resource management term is appearing.

Although Table 4. was not moved the font of numbers was enlarged.

Best regards,

Authors

Reviewer 2 Report

The creation of health 4.0 has a significant importance in the Management of healthcare system and had direct impact on the contributions of people working in healthcare entities. Thus the evolvement of HRM practice is a major necessity.

The analysis and evaluation of HRM presented in this paper is a good action today in our world rising the awareness towards the healthcare system, however I would have preferred to see some concrete examples making this manuscript more lively.

Another notice is to add a colorful presenting the values elaborated through the development of this HRM analysis in the results section.

I believe that the readers, especially those who are not directly related to the field, would be more attracted to the paper through some graphical elaboration for the coming out of the paper.

One of the results suggested is to add a new profession called HR partner in healthcare, it would be interesting to lighten if hospitals are read for such enhancements, if they are aware of the new obligations of health 4.0 and their readiness to interact.

Author Response

Dear Sir or Madam,

Thank you for in-depth review of our paper. The suggestion for improvements were very useful and majority of them were engaged in order to improve the paper. Let us refer to the feedback:

S1. The analysis and evaluation of HRM presented in this paper is a good action today in our world rising the awareness towards the healthcare system, however I would have preferred to see some concrete examples making this manuscript more lively.

Because this paper is based on quantitative data we find is very hard to add case studies. However, this suggestion motivates us to perform such the study in the future.

S2. Another notice is to add a colorful presenting the values elaborated through the development of this HRM analysis in the results section. I believe that the readers, especially those who are not directly related to the field, would be more attracted to the paper through some graphical elaboration for the coming out of the paper.

Due to suggestions of other reviewers new tables were added. Unfortunately we could not find a way to convert a graphic presentation of data in tables into other forms of presentation. However, Table 4 was graphically amended.

Best regards,

Authors

Reviewer 3 Report

Thank you for providing me with the opportunity of reviewing this paper. I will provide some comments targeting the most pressing issues. I hope that authors find my comments constructive and that they will be able to use them in improving this paper.

In this manuscript, the authors focus on the issue of human resource management in context of medical professionals and Healthcare 4.0 concept.

The authors stated that the purpose of this article is to "analyze and evaluate the involvement and roles of the key HRM actors, such as line managers and HR specialists, within hospitals”. The authors further state that (lines 81-83) "conclusions drawn from the analysis will enable the identification of the potential of the hospitals to effectively encounter Healthcare 4.0 requirements".

In the manuscript title, as well as in the introduction and in the abstract, the authors emphasize that the research is focused on HRM in the context of introducing the concept of Healthcare 4.0 in Polish hospitals. The authors smoothly justify the role of Healthcare 4.0. and combine it with the requirements of modern health care. However, in the empirical part of the study (survey) I did not find any direct connections with the threads of Healthcare 4.0., which were previously presented in the introduction (e.g. line 40-44). For this reason, I am not convinced whether such strong emphasis on Healthcare 4.0 is justified, both in the title and the purpose of the paper. In my opinion, this article is strongly oriented on human resources management, however, connections of the empirical part with sustainable development and Industry / Healthcare 4.0. are small. They are only the background of the conducted research. The article also lacks research hypotheses referring to the purpose of the article and embedded in the literature.

The empirical material was collected on a sufficient sample. The study base on primary data, which I consider to be the strength of this article. The procedure of gathering empirical data was generally correctly described, but there is no information about when the survey was conducted. This can be important in the context of a dynamically changing socio-economic environment. There is also no information on the implementation of Healthcare 4.0 solutions in the studied hospitals. The authors declare that the study "was designed based on an extensive literature review and discussions among researchers" (line 159). However, at this point I would expect specific references. I believe that the respondents were properly described and grouped.

I have objections about the statistical methods used in this research. The distribution analysis is one of the simplest approach. The Likert scale allows the use of more complex statistical methods. I believe that statements such as: "the level and area of engagement of line managers and HR managers differ depending on the point of view of the group of respondents" (lines 266-268) should be supported by the results of statistical tests (it should be clarified whether is the relationship statistically significant). For example, the authors can use the non-parametric Kruskal-Wallis test and the post-hoc multiple comparison which are available in the Statistica package. I believe that the extension of statistical analysis is required.

I also suggest changing the structure of the article a bit. The combination of Results and Discussion instead of Discussion and Conclusions seems more reasonable to me, especially that in the current version of the paper lines 266-278 are the result analysis and no discussion.

I encourage the authors to broaden the discussion, in particular to compare the results to similar studies from other countries. Although the authors state that their study has universal features, it is not obvious from the reader's point of view. The research was conducted in one country, whose health service has its own specificity.

I also wonder if the study has any limitations? The authors did not address this aspect in the conclusions.

Finally, I would like to emphasize that the article has been very well edited and adapted to the journal requirements. I just wonder if table 4 could be part of the appendix in a more readable form.

Author Response

Dear Sir or Madam,

Thank you for in-depth review of our paper. The suggestion for improvements were very useful and majority of them were engaged in order to improve the paper. Let us refer to the feedback:

S1. The authors stated that the purpose of this article is to "analyze and evaluate the involvement and roles of the key HRM actors, such as line managers and HR specialists, within hospitals”. The authors further state that (lines 81-83) "conclusions drawn from the analysis will enable the identification of the potential of the hospitals to effectively encounter Healthcare 4.0 requirements". In the manuscript title, as well as in the introduction and in the abstract, the authors emphasize that the research is focused on HRM in the context of introducing the concept of Healthcare 4.0 in Polish hospitals. The authors smoothly justify the role of Healthcare 4.0. and combine it with the requirements of modern health care. However, in the empirical part of the study (survey) I did not find any direct connections with the threads of Healthcare 4.0., which were previously presented in the introduction (e.g. line 40-44). For this reason, I am not convinced whether such strong emphasis on Healthcare 4.0 is justified, both in the title and the purpose of the paper. In my opinion, this article is strongly oriented on human resources management, however, connections of the empirical part with sustainable development and Industry / Healthcare 4.0. are small. They are only the background of the conducted research. The article also lacks research hypotheses referring to the purpose of the article and embedded in the literature.

It has been rightly noted that the purpose of the article is to determine the role and importance of key HRM actors, such as a line manager and personnel department in hospitals (line 81-83). Such the definition of aim arose from an assumption that solutions concerning HR organizational structure in hospitals to be supportive in Heathcare 4.0 solutions implementation should reflect these from business oriented organizations (commonly described in the HRM literature). In addition, on the one hand, by favoring the reviewer's observation, and on the other explaining the structure of the article, we would like to note that the authors' intention was not to identify the relationship between the implementation of solutions appropriate for Healthcare 4.0 and the organizational aspect of HRM in Polish hospitals. The intention was, as indicated, to determine the actual state-of-art of examined reality. The theoretical content on Healthcare 4.0 was only intended to indicate the convergence of trends in the development of healthcare with other sectors of the economy and the spectrum of challenges associated with the implementation of innovative solutions related to Healthcare 4.0. The attention was focused on the organizational aspect because such solutions either stimulate or inhibit the implementation of changes. Therefore, attempts were made to establish a kind of 'readiness' of hospitals to absorb Healthcare 4.0 solutions. The hypothesis was not defined deliberately because the research was exploratory. This nature of research was determined by the identified research gap.

S2. The empirical material was collected on a sufficient sample. The study base on primary data, which I consider to be the strength of this article. The procedure of gathering empirical data was generally correctly described, but there is no information about when the survey was conducted. This can be important in the context of a dynamically changing socio-economic environment. There is also no information on the implementation of Healthcare 4.0 solutions in the studied hospitals. The authors declare that the study "was designed based on an extensive literature review and discussions among researchers" (line 159). However, at this point I would expect specific references. I believe that the respondents were properly described and grouped.

The study was performed in 2017. The information was added to the paper. Although the study is not quit recent it is still actual since changes in HRM in Polish hospitals are rare. This function de facto is basically totally abandoned due to financial challenges of the systems. Authors via various publications tend to bring the healthcare managers attention to this aspect of management.

As previously explained the intention of the paper was to focus mainly on HRM organizational aspect. Healthcare 4.0 solutions were not studied. 

It is extremely hard to point out the set of specific references that were used for the questionnaire creation due to the fact that the final version of it was largely influenced by practical knowledge of the researchers engaged in the research project. However, in order to meet the reviewer's expectations, we have pointed out several dominant positions of the literature that have become an inspiration for questions related to the issues analyzed in the paper.

S3. I have objections about the statistical methods used in this research. The distribution analysis is one of the simplest approach. The Likert scale allows the use of more complex statistical methods. I believe that statements such as: "the level and area of engagement of line managers and HR managers differ depending on the point of view of the group of respondents" (lines 266-268) should be supported by the results of statistical tests (it should be clarified whether is the relationship statistically significant). For example, the authors can use the non-parametric Kruskal-Wallis test and the post-hoc multiple comparison which are available in the Statistica package. I believe that the extension of statistical analysis is required.

This suggestion was extremely important and priceless for the paper improvement. As suggested more sophisticated statistical analysis was performed. In addition, not only Kruskal-Wallis test, but also U-Mann Whitney test was performed. Results of the analysis were incorporated in the text (also in the form of additional tables) and discussed.

S4. I also suggest changing the structure of the article a bit. The combination of Results and Discussion instead of Discussion and Conclusions seems more reasonable to me, especially that in the current version of the paper lines 266-278 are the result analysis and no discussion.

I encourage the authors to broaden the discussion, in particular to compare the results to similar studies from other countries. Although the authors state that their study has universal features, it is not obvious from the reader's point of view. The research was conducted in one country, whose health service has its own specificity.

I also wonder if the study has any limitations? The authors did not address this aspect in the conclusions.

Fraises which are more results then discussion oriented were deleted. Majority of the part ‘Discussion and conclusions’ was rewritten. Unfortunately we were not able to identify limitations due to universal aspect of HR management in hospitals independent from healthcare system construction. However, we identified the future areas of investigations.

Once again thank you for this developmental review, which largely influenced the final version.

Best regards,

Authors

Reviewer 4 Report

Based on the Health 4.0, there are two ways for HRM: throughout various human resource management practices; and simultaneously, by optimal solutions that consider the organizational aspects of HRM, in particular a set of roles and the engagement of critical HR actors such as HR specialists and line managers.  It is a very interesting research idea; however, here are some suggestions for the authors.

  1. Introduction
  2. The paragraphs in the Introduction Section need a link to the content in order to read smoothly.
  3. Research gaps need to be clarified instead of stating requirements for Health 4.0.
  4. Research contributions need to be clearly stated.
  5. Based on the current, the health 4.0 and HRM is not linked.

  1. Literature review
  2. The objective of this paper is to find optimal solutions that consider the organizational aspects of HRM. Therefore, the related papers should be found and compared in order to show the research gap.

  1. Material and methods
  2. what is the analysis method? How is the survey process conducted?

  1. Conclusion
  2. This section is suggested to rewrite focusing in the main contributions and future research of the study.

Author Response

Dear Sir or Madam,

Thank you for in-depth review of our paper. The suggestion for improvements were very useful and majority of them were engaged in order to improve the paper. Let us refer to the feedback:

S1. Introduction. The paragraphs in the Introduction Section need a link to the content in order to read smoothly. Research gaps need to be clarified instead of stating requirements for Health 4.0. Research contributions need to be clearly stated. Based on the current, the health 4.0 and HRM is not linked.

In the introduction the following sentence brings readers into the main area of investigation:

In-depth analysis of Healthcare 4.0 implications reveals the fact that the effects of the revolution are progressing in two different ways – in both medicine itself and in the management of healthcare organizations; this is especially true in hospitals, which create the foundation of the healthcare system. Both of these require an appropriate level of human resource management (HRM) [17].

S2. Literature review. The objective of this paper is to find optimal solutions that consider the organizational aspects of HRM. Therefore, the related papers should be found and compared in order to show the research gap.

Quoting:

… the purpose of the paper is to analyze and evaluate the involvement and roles of the key HRM actors, such as line managers and HR specialists, within hospitals. Furthermore, conclusions drawn from the analysis will enable the identification of the potential of the hospitals to effectively encounter Healthcare 4.0 requirements.

The appropriate literature review was performed within point 2. That is “The main actors of human resource management in hospitals: literature review”.

S3. Material and methods. What is the analysis method? How is the survey process conducted?

In the first two paragraphs of the point the methodology of the research is described.

S4. Conclusion

This section is suggested to rewrite focusing in the main contributions and future research of the study.

The section has been rewritten. Authors developed part connected with contribution of the research and also future areas of investigations were pointed out.

Best regards,

Authors

Round 2

Reviewer 3 Report

The author(s) incorporated moste of the suggested changes and recommendations and provided proper explanations. Therefore, the manuscript is worth publishing. Hence, I endorse paper acceptance.

Reviewer 4 Report

All the questions are addressed.